# Specific IgE and Basophil Activation Test by Microarray: A Promising Tool for Diagnosis of Platinum Compound Hypersensitivity Reactions

**DOI:** 10.3390/ijms25073890

**Published:** 2024-03-31

**Authors:** Carlos Fernández-Lozano, Claudia Geraldine Rita, Alicia Barra-Castro, Belén de la Hoz Caballer, Ernesto Roldán, Cristina Pueyo López, Javier Martinez-Botas, María Pilar Berges-Gimeno

**Affiliations:** 1Biochemistry-Research Department, Hospital Universitario Ramón y Cajal, Instituto Ramón y Cajal de Investigación Sanitaria, Carretera de Colmenar Km 9, 28034 Madrid, Spain; carlitos4mb@hotmail.com; 2Allergology Department, Hospital Universitario Ramón y Cajal, Instituto Ramón y Cajal de Investigación Sanitaria, Carretera de Colmenar Km 9, 28034 Madrid, Spain; alicia.barra.castro@gmail.com (A.B.-C.); bhozcaballer@gmail.com (B.d.l.H.C.); 3Immunology Service, Hospital Universitario Ramón y Cajal, Instituto Ramón y Cajal de Investigación Sanitaria, Carretera de Colmenar Km 9, 28034 Madrid, Spain; claudiaritag@gmail.com (C.G.R.); ernesto.roldan@salud.madrid.org (E.R.); 4Department of Medicine and Health Sciences, Alcalá University, 28801 Madrid, Spain; 5Pharmacy Department, Hospital Universitario Ramón y Cajal, Instituto Ramón y Cajal de Investigación Sanitaria, Carretera de Colmenar Km 9, 28034 Madrid, Spain; cristina.pueyo@salud.madrid.org

**Keywords:** basophil activation test, drug allergies, hypersensitivity reaction, microarray, platinum compounds

## Abstract

Drug hypersensitivity reactions (DHRs) to platinum-based compounds (PCs) are on the rise, and their personalized and safe management is essential to enable first-line treatment for these cancer patients. This study aimed to evaluate the usefulness of the basophil activation test by flow cytometry (BAT-FC) and the newly developed sIgE-microarray and BAT-microarray in diagnosing IgE-mediated hypersensitivity reactions to PCs. A total of 24 patients with DHRs to PCs (20 oxaliplatin and four carboplatin) were evaluated: thirteen patients were diagnosed as allergic with positive skin tests (STs) or drug provocation tests (DPTs), six patients were diagnosed as non-allergic with negative STs and DPTs, and five patients were classified as suspected allergic because DPTs could not be performed. In addition, four carboplatin-tolerant patients were included as controls. The BAT-FC was positive in 2 of 13 allergic patients, with a sensitivity of 15.4% and specificity of 100%. However, the sIgE- and BAT-microarray were positive in 11 of 13 DHR patients, giving a sensitivity of over 84.6% and a specificity of 90%. Except for one patient, all samples from the non-allergic and control groups were negative for sIgE- and BAT-microarray. Our experience indicated that the sIgE- and BAT-microarray could be helpful in the endophenotyping of IgE-mediated hypersensitivity reactions to PCs and may provide an advance in decision making for drug provocation testing.

## 1. Introduction

Platinum compounds (PCs) are drugs used to treat a wide variety of colorectal, gynecological, and pulmonary tumors. The incidence of drug hypersensitivity reactions (DHRs) with these drugs is directly related to the number of drug exposures (up to 40% with carboplatin after seven treatment cycles and 15% with oxaliplatin). The most frequent phenotype of DHRs to PCs is Type 1 (IgE mediated). Cytokine release reactions, mixed reactions, and, in isolated cases, type IV reactions have also been described [1,2].

PC skin tests (STs) are available for diagnosis and risk assessment for severe reactions. The sensitivity of STs varies between studies (26–100% for oxaliplatin and around 80% for carboplatin), and they have a high specificity (91–100%, for both) [2,3,4].

The gold standard of diagnosis for PC hypersensitivity is the drug provocation test (DPT). A few groups of authors have performed DPTs to rule out or confirm hypersensitivity to PCs. The largest published study of a DPT with PCs showed that 46% (43 of 93) of all DPTs were negative and the number of unnecessary desensitizations was reduced [3]. Nevertheless, the DPT is a high-risk technique that requires close medical surveillance by personnel with expertise in drug allergies in adequate safety settings [2,3].

The basophil activation test by flow cytometry (BAT-FC) measures the expression of activation markers on the basophil surface [5]. The BAT-FC has been suggested as a promising predictor of severe DHRs in platinum-treated patients and the risk of reactions during desensitization [6,7,8]. However, the number of patients in these studies has been small, and this tool has not yet been validated. On the other hand, specific IgE (sIgE) combined with STs seems to be a good predictor of platinum allergy, although it seems to have a reasonable specificity (75–100%) but low sensitivity (34–75%) [3,9].

Microarray technology has allowed for the simultaneous assessment of sIgE levels of hundreds of allergens, enabling the development of component-resolved diagnostics in an affordable way [10,11]. Humanized rat basophilic leukemia cell lines have been proposed as a suitable replacement for peripheral human basophils for detecting allergen- sIgE reactivity [12,13]. The combination of humanized rat basophil cell lines and microarray technology combines high-throughput screening and cellular activation rather than only sIgE binding [14,15]. The BAT-FC has some limitations, including the need to process fresh samples, a small percentage of patients with non-responsive basophils, and it requires a larger amount of blood than other techniques, especially when patients are treated with cytostatics that reduce leukocyte counts, such as in oncological treatments.

Until now, the diagnosis of hypersensitivity reactions to antineoplastic drugs has been based on a detailed clinical history and skin testing. In addition, DPTs can be performed in some cases but are only available in a limited number of centers. The development of in vitro tests to optimize the diagnosis of these reactions is essential for patients who may be receiving first-line treatment for their cancer.

The aim of this study was to explore the usefulness of in vitro techniques (the BAT-FC, BAT-microarray, and sIgE-microarray) in a group of patients presenting hypersensitivity reactions to PCs. We compared these assay results with the clinical diagnosis (STs and, if possible, DPTs).

## 2. Results

### 2.1. Characteristics of the Patients

Twenty-eight patients (twenty-two treated with oxaliplatin and six with carboplatin) gave their consent to participate in the study. Only 24 patients (20 treated with oxaliplatin and 4 with carboplatin) were finally evaluated as the remaining patients withdrew from the study (Figure 1). We also included four patients treated with carboplatin who had not experienced a DHR (as non-allergic control). The baseline characteristics of the patients are presented in Table 1. The patients’ ages ranged from 34 to 82 years (mean 63 years), and 28 patients were being treated for a malignancy (19 colorectal, 5 genital, 1 lung, and 1 breast). An amount of 15 of 24 (62%) patients suffered a moderate or severe initial reaction (corresponding to Brown grade 2–3 and Ramón y Cajal University Hospital _-RCUH- grade 2–4), and 9 of 24 (38%) presented a mild initial reaction (corresponding to Brown grade 1 and RCUH grade 1).

### 2.2. Allergological Work-Up Outcomes

The results of the allergological study are shown in Table 2. Of the 24 patients, 11 had a positive ST (45.8%), and 13 were negative (54.2%) (Figure 1). The negative patients underwent a DPT; two had a positive DPTs (25%), and six were negative (75%). DPTs were not possible in five patients; one had a severe reaction (patient 5), and four patients refused to undergo a DPT (patients 13, 14, 22, and 24); all were classified as suspected allergic (Figure 1). Of the two patients with a positive DPT, patient 3 suffered a mild reaction (local urticarial and palmar pruritus and needed intravenous antihistamines and corticosteroids). The other patient (#4) presented a moderate reaction (severe back pain, headache, and hypertension controlled by antihistamines, corticosteroids, and paracetamol administered intravenously). All of them were subjected to our “restart protocol” for positive patients and showed good tolerance to oxaliplatin.

### 2.3. Basophil Activation Test by Flow Cytometry (BAT-FC)

The BAT-FC was performed on 23 of the 24 study patients and 2 of the 4 control samples. Only two patients in the allergic group had a positive BAT-FC. Patient 6 had CD63^+^ basophil percentages of 24.8%, 26.5%, and 35.4% at oxaliplatin concentrations of 50, 125, and 250 µg/mL, respectively (Figure 2). In this case, the SI of CD203c was >2 for the two highest drug concentrations (125 and 250 µg/mL). In addition, patient 21 showed an increase in CD63+ expression at carboplatin concentrations of 125 µg/mL (CD63+: 8.2%) and 250 µg/mL (CD63+: 9.2%); the SI of CD203c remained negative at all concentrations tested in this patient.

The remaining allergic patients had a negative BAT-FC. However, two were excluded from the analysis, one because of a higher basal activation of basophils and the other because of non-stimulation of the positive control. CD203c could not be stained in four patients. No positive BAT-FC was found in the control or non-allergic groups. The overall sensitivity and specificity of the BAT-FC, including both drugs, were 15.4% and 100%, respectively. Therefore, the low sensitivity of the BAT-FC made this technique ineffective to classify patients in our study.

### 2.4. sIgE-Microarrays Immunoassay

To study the oxaliplatin- and carboplatin-sIgE, we developed a microarray containing the drugs complexed with different amounts of albumin (Figure 3). We tested three molar ratios of drug/albumin (1:10, 1:100, and 1:1000). The best performing ratio for oxaliplatin was 1:100 (Figure 4A and Appendix A). For carboplatin, the three-molar ratio had good performance (Figure 4B and Appendix A). In order to compare the data, we selected the higher values of the sIgE 1:100 molar ratio (Table 2).

As shown in Figure 4A, in the oxaliplatin-allergic group, 10 out of 11 patients (90.9%) had positive oxaliplatin sIgE (*z*-score > 3). In most cases, there was a dose effect, and the amount of IgE increased by increasing drug concentration. Of the patients with suspected oxaliplatin allergies, patient 13 was negative, patient 5 had moderate levels, and patient 14 had elevated IgE levels to the drug. All samples in the non-allergic group (except patient 19) were negative for sIgE.

Both patients with confirmed carboplatin allergies had elevated IgE levels (Figure 4B). Of the two patients with a suspected carboplatin allergy, 24 had high IgE levels, and 22 were slightly above the cut-off value (*z*-score > 3). The overall sensitivity and specificity of the sIgE-microarrays, including both drugs, were 92.3% and 90%, respectively. In conclusion, the sIgE-microarray immunoassay showed excellent performance in the classification of patients with IgE-mediated hypersensitivity reactions to PCs.

### 2.5. Basophil Activation Test on Microarray Support (BAT-Microarray)

A BAT-microarray was performed on all patients included in the study. A microarray containing the tested drugs complexed with different amounts of albumin (1:10, 1:100, and 1:1000 molar ratios) was incubated with RBL-30/25 cells previously sensitized with the patients’ serum. Cell activation was assessed by the surface expression of CD63 (Figure 3). For both oxaliplatin and carboplatin, the best performing molar ratio was 1:100 (Figure 5 and Appendix A). As shown in Figure 5A, in the oxaliplatin-allergic group, 9 out of 11 (81,8%) had a positive BAT-microarray at the 1:100 molar ratio. Of the patients with suspected oxaliplatin allergies, patients 5 and 14 had a positive BAT-microarray, and patient 13 was negative. All samples in the non-allergic group (except patient 19) had a negative BAT-microarray (Figure 5A).

On the other hand, both patients with a confirmed allergy to carboplatin had a positive BAT-microarray in the 1:100 molar ratio (Figure 5B). Of the two patients with suspected carboplatin allergies, 24 had a positive BAT-microarray, and 22 had a negative BAT-microarray. All samples in the control group had a negative BAT-microarray (Figure 5B). The overall sensitivity and specificity of the BAT-microarray were 84.6% and 90%, respectively. According to our results, both sIgE- and BAT-microarray showed excellent performance and could improve the endophenotyping of IgE-mediated hypersensitivity reactions to platinum compounds.

## 3. Discussion

The DPT is considered the gold standard in the diagnosis of drug allergies, and its use in allergy clinical practice helps to establish or exclude hypersensitivity. However, the DPT is a high-risk procedure, and the validation of other diagnostic tools, such as the BAT-FC, is essential to reduce the risk of the DPT. No studies have compared the results of the BAT-FC with those of the DPT in patients with DHRs to platinum compounds. In the present study, we evaluated the utility of the BAT-FC and, for the first time, the sIgE- and BAT-microarray as diagnostic tools for DHRs to PCs. Our results revealed that the BAT-FC sensitivity was low at only 14.4%, compared to those of the sIgE- and BAT-microarray, which were higher than 84%.

Interestingly, we found four patients with non-immediate reactions. In three of these patients, the STs and DPTs were negative (in patient 13, no challenge could be performed), and, as expected, the IgE and BAT were negative. In the fourth patient, the sIgE- and BAT-microarray were positive, which could indicate a mixed endotype.

In addition, one patient diagnosed with a DHR did not exhibit a sIgE and yielded negative results on the BAT-microarray, potentially indicating an IgE-independent mechanism such as Mas-related G protein–coupled receptor X2 (MRGPRX2) [16], which could not be further investigated.

It is well known that BAT-FC sensitivity for diagnosing IgE-mediated hypersensitivity reactions differs according to the drug tested [17]. Over the last years, the potential value of the BAT-FC with different basophil activation markers (CD203c and CD63) in PC allergies has been investigated [6,7,8]. Although the BAT-FC is a good ex vivo technique in allergy assessment, it still lacks standardization. Each center employs different allergen preparations, dilutions, incubation times, and flow cytometry essential parameters, such as the data analysis, gating strategies, and positivity thresholds established in different laboratories, that could influence the BAT-FC results [18,19]. In this regard, in the present study, the positivity threshold was chosen based on previous experiences with other non-chemotherapeutic drugs [20,21]. However, compared with other studies, our cut-off was more restrictive than those published by other groups [6,7,8].

The BAT-FC is a minimally invasive, safer, and less expensive tool than the DPT in diagnosing DHRs to platinum compounds. However, the BAT-FC has some limitations, such as the completion time, as the sample must be investigated within 24 h [19]; the test requires qualified personnel and needs validation in cohorts of adequate statistical power and in patients well-characterized (diagnosed as positive or negative after a protocol including STs and DPTs). The time between the reaction and the test is an important aspect. In cancer patients, the interval to perform this diagnostic procedure is usually two weeks, a short interval that can interfere with BAT-FC results (basophils hyporeactivity response or anergy) [22].

Due to the low frequency of the human peripheral blood basophils (less than 1%) [14] and the difficulty of their purification [23], several rat basophilic leukemia (RBL) cell lines transfected with human FcεRI have been developed to be used as a diagnostic tool for the detection of allergen-sIgE and as a surrogate for patient basophils [12,24]. Among them, RBL-30/25 and RBL SX-38 cell lines presented an optimal IgE-mediated degranulation [19,24]. In the present work, we improved the protocol of the BAT-microarray by using 96-well polystyrene plates instead of glass slides that allowed for a good deposition of allergens and provided an appropriate environment for cell cultures. Our results showed that the BAT-microarray, with a sensitivity of 84.6% and specificity of 90%, had a better diagnostic performance than the BAT-FC, with a sensitivity of 15.4% and specificity of 100%.

The main limitation of RBL cells expressing the high-affinity IgE receptor is that, being rat cells, it is not known whether they can respond to the other molecules involved in the allergic response present in the patient’s serum (cytokines, IgG4, IgG4 receptor, etc.) and thus resemble the activation of the patient’s basophils or merely sensitization related to the presence of IgE. In the present work, we found some differences between the sIgE- and BAT-microarray. For instance, the serum of patients 7 and 10 had high levels of IgE but a limited ability to induce basophil activation. On the other hand, patients 6, 8, and 21 presented a stronger basophil activation than would be expected based on their sIgE levels. This suggested that the two assays were not identical and could provide complementary information, as is the case with sIgE and basophil activation with the patient’s cells. However, this aspect needs further study.

Oxaliplatin- and carboplatin-sIgE was also determined in all patients included in this study by microarray immunoassays. In previous studies, the determination of sIgE to platinum compounds has also been investigated as a diagnostic method using the ThermoFisher InmunoCAP technique [9]. Our group, in a previous study, showed that the sensitivity for oxaliplatin-sIgE was 34% and the specificity was 90%. However, Caiado J et al. reported lower values of specificity of oxaliplatin-sIgE (75%) and higher sensitivity values (75%). This work also showed sensitivity (59%) and specificity (100%) for carboplatin-sIgE. Our results showed that sIgE to oxaliplatin by microarray immunoassay correlated with the results of the STs and DPTs with a 90.9% sensitivity and 83.3% specificity. Similar results, albeit with fewer samples, were observed with carboplatin. Therefore, sIgE by microarray and the BAT-microarray could be complementary tools for diagnosing hypersensitivity reactions to chemotherapeutic agents such as oxaliplatin and carboplatin.

The implementation of DPTs in diagnostic protocols helps to validate diagnostics tools. For the first time, we evaluated the roles of the sIgE- and BAT-microarray in the most extensive series of patients with hypersensitivity to oxaliplatin reported to date, including DPTs in the diagnosis. Our results showed an improvement in sIgE determination and the BAT-microarray, which could help identify endophenotypes to hypersensitivity reactions to platinum compounds. These in vitro diagnostic tools could clinically guide in identifying candidates for DPTs and could be considered a safe alternative in cases of severe reactions or high-risk patients. Further, more extensive multicenter studies are needed to evaluate the BAT-microarray in diagnosing platinum compound DHRs.

## 4. Materials and Methods

### 4.1. Patient Population and Study Design

We carried out a prospective, observational, longitudinal study with consecutive patients who suffered from DHRs due to PC and were referred to the Allergy Division’s Desensitization Program of RCUH between January 2019 and June 2022. The study was conducted in accordance with the Declaration of Helsinki, approved by the RCUH Ethics Committee (institutional register number: 268/18), and written informed consent was obtained from all participants.

The diagnostic protocol was based on a detailed clinical history, STs, basal serum tryptase determination, risk assessment, and DPTs, when the appropriate criteria were met [3].

Initial DHRs were classified as immediate and non-immediate and according to severity as in previous publications [3,25,26].

We considered high-risk patients those who met any of the following criteria: previous severe (life-threatening) reaction, comorbidities such as uncontrolled lung disease with a forced expiratory volume < 1 L, severe systemic illness, or unavoidable use of beta-blockers.

STs were performed following the safe handling of antineoplastic requirements. The skin prick test (SPT) concentration for oxaliplatin was 5 mg/mL, and the intradermal test (IDT) concentrations were 0.5 and 5 mg/mL. Carboplatin STs were performed at 10 mg/mL for the SPT and 1 mg/mL, and 1 mg/mL for the IDT. Positive and negative controls were carried out with saline and histamine, respectively. A positive reaction was defined according to the European Network for Drug Allergy guideline [27]. Standard ST protocols use a stepwise SPT followed by an IDT if the former was negative.

DPTs were carried out in the medical intensive care unit, following the same detailed recommendations described in other articles by our group [3]. It was considered positive when it reproduced the original symptoms or showed an objective DHR. Serum tryptase levels were determined during the reaction. The final diagnosis of a DHR was defined as allergic based on a positive ST or positive drug provocation tests to PCs. Patients with negative DPT results were considered non-allergic. A DPT was not performed in high-risk patients or patients who did not consent; these patients were considered suspected allergic. Patients were on multi-drug regimens and were separately assessed if more than one drug was suspected to avoid misdiagnosis.

### 4.2. Basophil Activation Test by Flow Cytometry (BAT-FC)

An amount of 200 µL of whole blood was preincubated for 10 min at 37 °C with 40 µL of stimulation buffer, containing fetal bovine serum and human IL-3 (9 ng/mL final concentration, Pharmingen, San Diego, CA, USA). Subsequently, the samples were incubated at 37 °C for 30 min with four concentrations of oxaliplatin or carboplatin (5, 50, 125, and 250 µg/mL). Anti-IgE antibodies (BD Biosciences, Franklin Lakes, NJ, USA) and PBS were the positive and negative controls. Samples were incubated on ice for 5 min to stop the reaction and stained in the dark for 30 min with 20 µL of the antibody cocktail: anti-CD63-FITC, anti-CD123-PE, and anti-HLA-DR-PerCP (FastImmune™; BD Biosciences). In some experiments, 5 µL of CD203c-BV421 (BD Biosciences) was also added. Red cells were lysed with FACS™ Lysing Solution. Samples were analyzed immediately on a FACS Canto II flow cytometer (BD Biosciences) until each tube volume was exhausted or at least 500 basophil events were obtained. An example of the gating strategy is shown in Figure 2A.

CD63 activation was measured as the percentage of CD63+ cells after subtracting the percentage of basally activated basophils in the negative control. The CD203c stimulation index (SI) was calculated as the ratio of the mean fluorescence intensity (MFI) of CD203c between stimulated and unstimulated basophils. A BAT-FC was considered positive when the CD203c SI was >2 and/or the percentage of the CD63+ basophil was >5%. The BAT-FC results were considered invalid when the proportion of CD63+ basophils in the positive control was <5% or when the basal activation of basophils was >2%.

### 4.3. Microarray Printing

Oxaliplatin and carboplatin conjugates were prepared similarly to that described by Caiado [9]. Oxaliplatin and carboplatin were mixed with different amounts of human albumin to obtain molar ratios of 1:10, 1:100, and 1:1000. The solutions were incubated for 18 h at room temperature with gentle shaking. The unbound drug was removed using Slide-A-Lyzer^®^ MINI Dialysis Device columns 10K MWCO (Thermo Scientific, Waltham, MA, USA). The solutions were diluted with PBS and Protein Printing Buffer (Arrayit Corporation, Sunnyvale, CA, USA) to obtain the following concentrations: 0.062, 0.125, 0.25, 0.5, and 1 µg/µL, except for the 1:1000 molar ratio where only the concentration of 0.4 µg/µL could be achieved. Biotinylated anti-rat IgG antibodies (0.1 µg/µL) (112–065-062, Jackson Immunoresearch, West Grove, PA, USA) were printed as a positive reference control. Anti-human IgE (2 µg/µL) (DIA HE1, BioPorto Diagnostics, Boston, MA, USA) was printed as a positive control. PBS, Angiotensin II octa peptide (1 µg/µL) (Sigma-Aldrich, St. Louis, MO, USA), poly-DL-alanine (1 µg/µL) (Sigma-Aldrich), and human serum albumin (HSA) (1, 0.2, and 0.04 µg/µL) (Grifols, Barcelona, Spain) were printed as negative controls. The microarray printing was performed as previously described using a sciFLEXARRAYER S3 piezoelectric spotter and a PDC 70 Type 2 (Scienion, Berlin, Germany). Each feature was printed in triplicate on glass slides for the IgE detection assay (sciCHIP Epoxy slides; Scienion) or 96-well microtiter plates for the BAT-microarray (sciPLEXPLATE Type 2; Scienion). After printing, the arrays were dried at RT overnight and stored in a desiccator. Before use, printed surfaces were washed twice with PBS and blocked with Protein Microarray Activation Buffer (PMAB, Scienion) for 1 h at RT (sIgE-microarrays) or 3 mg/mL HSA diluted in PBS (HSA/PBS) for 1.5 h at RT (BAT-microarray).

### 4.4. sIgE Microarray Immunoassay

After blocking, the microarray slides were washed twice with Protein Microarray Wash Buffer (PMWB, Scienion). Patient serum was diluted to 1:1 with Protein Microarray Reaction Buffer (PMRB, Scienion), incubated with the microarray for 2.5 h at 37 °C with gentle agitation, and washed 3 times with PMWB. The microarrays were then incubated with a biotinylated anti-human IgE antibody (clone G7–26, BD Pharmingen, Franklin Lakes, NJ, USA), diluted to 1:50 in PMRB for 2 h at 37 °C, and washed 3 times with PMWB. Finally, the microarrays were incubated with Cy3-streptavidin (PA43001; Cytiva, Marlborough, MA, USA), diluted to 1:500 in PMRB for 30 min at RT, washed 6 times with PMWB, and scanned using a Scan Array Express (Perkin Elmer, Waltham, MA, USA).

### 4.5. Basophil Activation Test on Microarray Support (BAT-Microarray)

The BAT-microarray was performed using rat basophil leukemia cells (RBL-30/25) transfected with the human FcεRIα receptor, produced at the Paul-Ehrlich-Institut (Langen, Germany; kindly donated by Stefan Vieths and Lothar Vogel) [28]. Cells were cultured in MEM (GIBCO)-10% fetal bovine serum (Thermo Scientific) at 37 °C in a 5% CO_2_ atmosphere.

RBL-30/25 cells were seeded (7 × 10^5^/300 µL) in a 12-well plate. After 8 h, cells were sensitized with 5% heat-inactivated serum of the patients for 16 h, transferred to a 96-well microtiter plate containing the microarray, and incubated for 2.5 h under slow agitation. Afterward, cells were washed 3 times with 1 mg/mL HSA/PBS (WB) for 10 min under slow agitation, fixed with 4% formaldehyde for 1 h, and washed 3 times with WB. After that, we added the biotinylated anti-rat CD63 antibodies (Clon REA444, Miltenyi Biotech, Bergisch Gladbach, Germany), diluted to 1:50 with WB for 30 min, and washed 3 times with WB. Finally, the cells were incubated for 30 min at RT with a Cy3-Streptavidin (PA43001; Cytiva), diluted to 1:500 with WB, washed 6 times with WB, and scanned using a sciREADER FL2 (Scienion).

### 4.6. Data Analysis

Microarray analysis was performed according to the method of Lin et al. [29]. An individual spot was considered positive if the standardized intensity of fluorescence (represented as the weighted average *z*-score) exceeded 3. Data analysis and presentation were performed using Microsoft Excel (https://www.microsoft.com/en-us/microsoft-365/excel accessed on 25 March 2024) and R (https://www.r-project.org/ accessed on 25 March 2024).

## Figures and Tables

**Figure 1 ijms-25-03890-f001:**
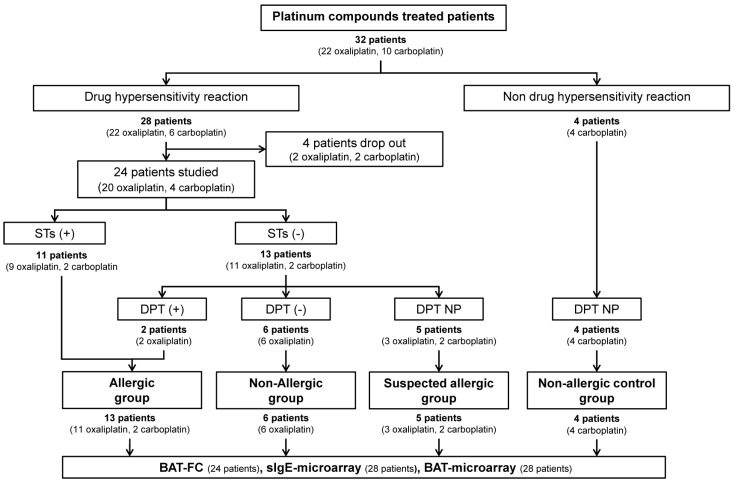
Schematic flow diagram of recruitment of platinum-compound-treated patients. STs, skin tests; DPT, drug provocation test; DPT NP, drug provocation test not performed; BAT-FC, basophil activation test by flow cytometry; sIgE-microarray, specific IgE by microarray; BAT-microarray, basophil activation test by microarray.

**Figure 2 ijms-25-03890-f002:**
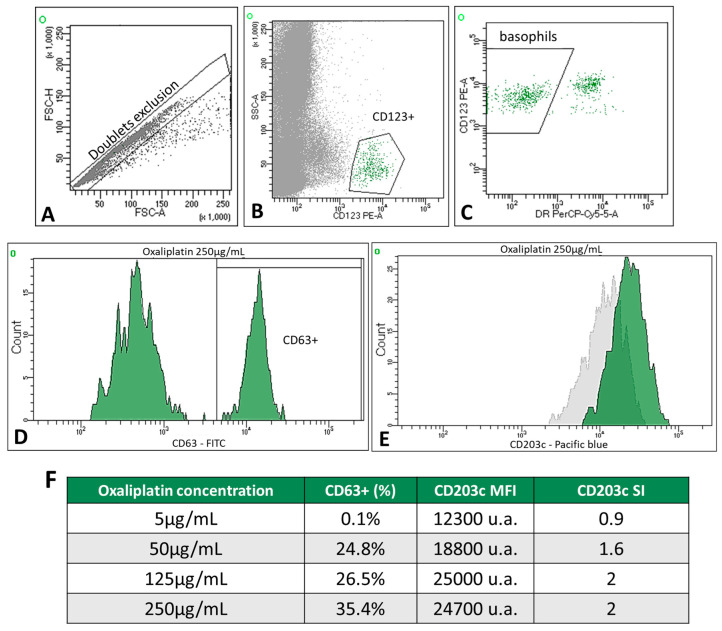
Example of a positive BAT-FC in patient 6. Gating strategy for basophils: (**A**) doublet exclusion, FSC-H vs. FSC-A; basophils were identified as low SSC-A and CD123^+^ (**B**) and CD123^+^ HLA-DR^neg^ cells (**C**). (**D**) The univariate histogram shows a positive BAT determined by an increased percentage of CD63 expression on the patient’s basophils with the highest concentration of oxaliplatin (250 µg/mL). (**E**) The univariate histogram (green) shows a positive BAT determined by increased mean fluorescence intensity (MFI) on the patient’s basophils at 250 µg/mL oxaliplatin; the gray histogram corresponds to the unstimulated cells. (**F**) Table showing the percentage of CD63+, MFI, and SI of CD203c in patient 6. Data were analyzed using FACSDiva software (verion 8.0.2).

**Figure 3 ijms-25-03890-f003:**
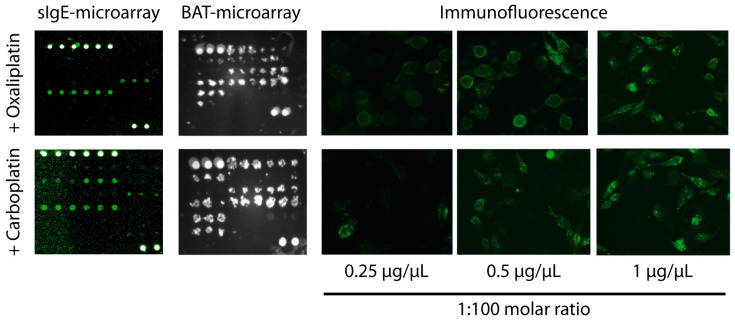
Examples of positive sIgE-microarray, BAT-microarray, and immunofluorescence detection of CD63 upon drug stimulation in patient 6, allergic to oxaliplatin, and patient 21, allergic to carboplatin. Cells were visualized under at ×100 magnification using a Olympus IX70 fluorescent microscope.

**Figure 4 ijms-25-03890-f004:**
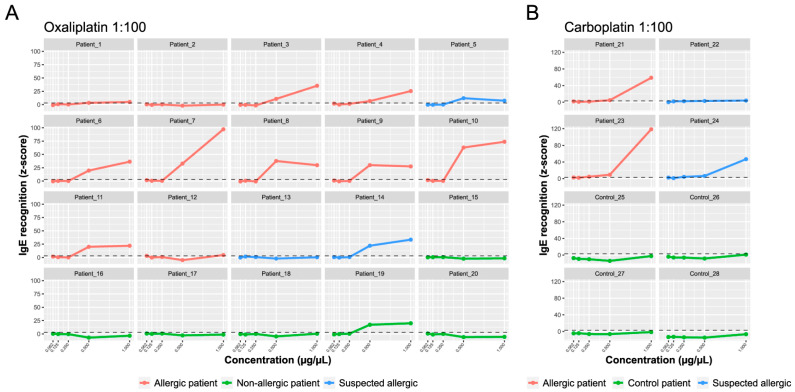
sIgE-microarrays. The *y*-axis shows the average IgE binding represented as average *z*-scores. The *x*-axis shows the drug concentration: oxaliplatin (**A**) and carboplatin (**B**) 1:100 molar ratio. The standardized fluorescence intensity represented as the average *z*-score was considered positive if it exceeded three (dotted line).

**Figure 5 ijms-25-03890-f005:**
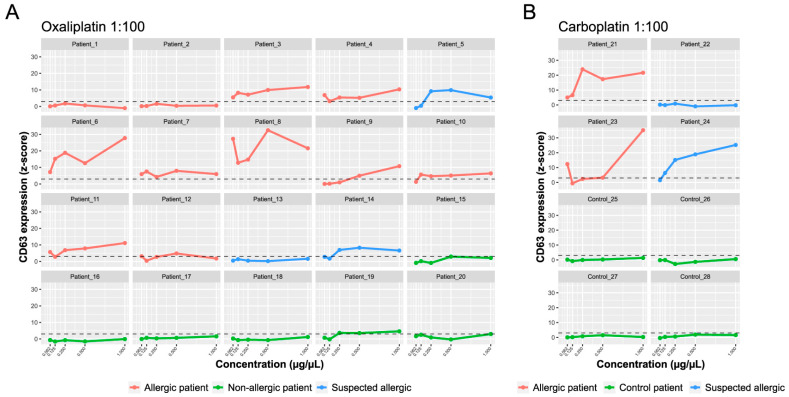
BAT-microarray immunoassay. The *y*-axis shows the average CD63 expression represented as average *z*-scores. The *x*-axis shows the drug concentration: oxaliplatin (**A**) and carboplatin (**B**) 1:100 molar ratio. The standardized fluorescence intensity represented as the average *z*-score was considered positive if it exceeded three (dotted line).

**Table 1 ijms-25-03890-t001:** General characteristics of patients and platinum compound hypersensitivity reactions and general characteristics of tolerant patients (control group).

Patient	Type of Tumor	Sex	Age (Years)	Drug	Grade of Initial Reaction Brown/HRYC	Lifetime Exposure (Previous Cycles)	Time to Index Reaction (min)	Symptoms	History of Atopic Disease/HR
1	Colorectal	F	68	Oxaliplatin	1.1	7	60	Urticaria, erythema, pruritus	No/No
2	Colorectal	M	69	Oxaliplatin	2.2	20	5	Urticaria, erythema, pruritus, nausea	No/No
3	Colon	M	76	Oxaliplatin	1.1	4	30	Erythema, pruritus, warm sensation	No/No
4	Sigmoid	F	63	Oxaliplatin	2.1–2	9	60	Warm, diaphoresis, chills	No/No
5	Colon	M	71	Oxaliplatin	3.3	10	120	Erythema, flushing, pruritus, desaturation	No/No
6	Gastric	F	53	Oxaliplatin	2.3	6	45	Erythema, flushing, pruritus, chest tightness	No/No
7	Rectal	F	64	Oxaliplatin	1.1	3	60	Erythema, flushing	No/No
8	Sigmoid and rectal	M	63	Oxaliplatin	3.3	9	70	Pruritus, maculopapular rash, desaturation	Yes/Yes
9	Colon	M	68	Oxaliplatin	3.3	3	60	Cough, desaturation, throat tightness	No/No
10	Colorectal	F	58	Oxaliplatin	3.4	3	60	Erythema, flushing, dyspnea, desaturation, syncope, confusion, dizziness	No/Yes
11	Colon	M	70	Oxaliplatin	3.3	5	20	Erythema, flushing, pruritus, desaturation	Yes/No
12	Sigmoid	F	71	Oxaliplatin	1.1	9	5	Localized urticaria, pruritus	No/No
13	Sigmoid	M	60	Oxaliplatin	2.2	1	120	Dysesthesias, dyspnea, dysphonia, uvula edema	Yes/Yes
14	Sigmoid	F	70	Oxaliplatin	1.1	3	60	Localized urticaria, pruritus	No/No
15	Colon	M	49	Oxaliplatin	1.1	8	70	Erythema, Dysesthesias	Yes/No
16	Sigmoid	F	68	Oxaliplatin	2.2	1	60	Dyspnea, dysphonia, tachycardia	No/No
17	Sigmoid	F	75	Oxaliplatin	2.3	1	120	Dyspnea, wheeze	Yes/No
18	Rectal	M	82	Oxaliplatin	1.1	22	480	Flushing	Yes/No
19	Colon	F	57	Oxaliplatin	2.3	1	60	Dysesthesias, dyspnea, throat tightness	No/No
20	Colorectal	F	58	Oxaliplatin	1.1	3	48	Pruritus, dysesthesia	No/No
21	Endometrial	F	57	Carboplatin	3.3	6	15	Dyspnea, desaturation, hypertension, chest pain	No/Yes
22	Ovarian	F	48	Carboplatin	2.2	15	Not Known	Localized urticaria, dyspnea	No/Yes
23	Lung	M	75	Carboplatin	3.3	7	60	Pruritus, dyspnea, desaturation	No/No
24	Ovarian	F	69	Carboplatin	1.1	9	15	Urticaria, hands and foot pruritus	No/No
25	Lung	M	73	Carboplatin	NR ^‡^	1	NA ^†^	NA ^†^	No/No
26	Vulvar	F	53	Carboplatin	NR ^‡^	2	NA ^†^	NA ^†^	Yes/Yes
27	Ovarian	F	43	Carboplatin	NR ^‡^	1	NA ^†^	NA ^†^	No/Yes
28	Breast	F	34	Carboplatin	NR ^‡^	4	NA ^†^	NA ^†^	No/Yes

^†^ NA: Not applicable, ^‡^ NR: Not registered.

**Table 2 ijms-25-03890-t002:** Allergological work-up outcome.

Patient	Drug	Skin Tests	DPT	Tryptase (Basal)	Tryptase (Reaction)	Total IgE	BAT-FC CD63	BAT-FC CD203c	sIgE-Microarray	BAT-Microarray
1	Oxaliplatin	+(IDT 0.5 mg/mL)	NP ^†^	6.67	NP ^†^	354	-	NP ^†^	4.96	ND ^‡^
2	Oxaliplatin	+(IDT 0.5 mg/ml	NP ^†^	7.57	NP ^†^	71	-	-	0.02	0.55
3	Oxaliplatin	-	+	<1	NP ^†^	4	-	-	35.39	11.75
4	Oxaliplatin	-	+	13.8	13.70	6	-	-	25.40	10.33
5	Oxaliplatin	-	NP ^†^	<1	NP ^†^	157	-	-	7.37	5.39
6	Oxaliplatin	+(IDT 5 mg/mL)	NP ^†^	1.03	NP ^†^	43	+	+	36.37	27.76
7	Oxaliplatin	+(IDT 0.5 mg/mL)	NP ^†^	7.34	NP ^†^	608	-	-	97.20	5.96
8	Oxaliplatin	+(SPT 0.5–5 mg/mL)	NP ^†^	6.69	NP ^†^	798	-	-	29.79	21.57
9	Oxaliplatin	+(IDT 5 mg/mL)	NP ^†^	5.09	NP ^†^	240	-	-	27.45	10.75
10	Oxaliplatin	+(IDT 0.005 mg/mL)	NP ^†^	7.49	NP ^†^	23	-	-	73.75	6.38
11	Oxaliplatin	+(IDT 0.5 mg/mL)	NP ^†^	6.56	NP ^†^	17	-	-	21.99	11.12
12	Oxaliplatin	+(IDT 0.5 mg/mL)	NP ^†^	6.06	NP ^†^	29	-	-	4.57	4.84 *
13	Oxaliplatin	-	NP ^†^	5.08	NP ^†^	90	-	-	0.38	1.61
14	Oxaliplatin	-	NP ^†^	5.81	NP ^†^	71	NP ^†^	NP ^†^	33.45	6.60
15	Oxaliplatin	-	-	6.32	NP ^†^	38	-	-	ND ^‡^	2.11
16	Oxaliplatin	-	-	3.6	NP ^†^	32	-	-	0.32	ND ^‡^
17	Oxaliplatin	-	-	4.37	NP ^†^	9	NV ^§^	-	ND	1.57
18	Oxaliplatin	-	-	3.02	NP ^†^	633	-	-	0.36	1.16
19	Oxaliplatin	-	-	3.41	NP ^†^	463	NV ^§^	NP ^†^	19.91	4.60
20	Oxaliplatin	-	-	7.32	NP ^†^	41	-	NP ^†^	ND ^‡^	2.90
21	Carboplatin	+(IDT 1 mg/mL)	NP ^†^	2.14	NP ^†^	743	+	-	58.76	21.62
22	Carboplatin	-	NP ^†^	4.48	NP ^†^	54	-	-	3.92	ND ^‡^
23	Carboplatin	+(IDT 1 mg/mL)	NP ^†^	7.41	NP ^†^	49	-	-	118.72	35.11
24	Carboplatin	-	NP ^†^	3.67	NP ^†^	5	-	-	46.76	25.21
25	Carboplatin	NP ^†^	NP ^†^	NP ^†^	NP ^†^	NP ^†^	-	-	ND ^‡^	1.33
26	Carboplatin	NP ^†^	NP ^†^	NP ^†^	NP ^†^	NP ^†^	-	-	1.03	0.56
27	Carboplatin	NP ^†^	NP ^†^	NP ^†^	NP ^†^	NP ^†^	NP ^†^	NP ^†^	ND ^‡^	0.26
28	Carboplatin	NP ^†^	NP ^†^	NP ^†^	NP ^†^	NP ^†^	NP ^†^	NP ^†^	ND ^‡^	1.59

^†^ Not Performed, ^‡^ Not detected, ^§^ Not valid. * Determination to Oxaliplatin 1:100 (0.5 µg/µL). IDT, intradermal test; SPT, skin prick test (mg/mL). Tryptase (µg/L), IgE total (kU/L), sIgE-microarrays (*z*-score), BAT-microarray (*z*-score). - means negative test, + means positive test.

## Data Availability

The raw data supporting the conclusions of this article will be made available by the authors on request.

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
