# Peer review of "Specific IgE and Basophil Activation Test by Microarray: A Promising Tool for Diagnosis of Platinum Compound Hypersensitivity Reactions"

_ijms, 2024, doi:10.3390/ijms25073890_

Round 1

Reviewer 1 Report

Comments and Suggestions for Authors

In this manuscript, the authors evaluated the value of basophil activation assay by flow cytometry and the newly developed sIgE-microarray and BAT-microarray in diagnosing IgE-medi-26 ated hypersensitivity reactions to platinum-based compounds. This study is of great significance for the detection of drug hypersensitivity reactions caused by platinum compounds (PCS), but there are still some questions and suggestions as follows:

1.     Add a section on the significance of your study at the end of the Introduction.

2.     Summarize the findings at the end of each section in the Results.

3.     Please explain the age of the participants and whether the underlying medical conditions are common.

4.     It is recommended that the content format in the table be centered.

5.     On line 305, what was the basis for choosing these three concentrations?

Author Response

Dear reviewer,

Thank you for your careful review of our paper and for the valuable comments, corrections, and suggestions that have helped us to improve the manuscript. Below, you will find the point-by-point response to your questions.

  1. Add a section on the significance of your study at the end of the Introduction.

Following your recommendation, we have introduced the following paragraph at the end of the introduction (line 74):

 “Until now, the diagnosis of hypersensitivity reactions to antineoplastic drugs has been based on a detailed clinical history and skin testing. In addition, DPT can be performed in some cases, but is only available in a limited number of centres. The development of in vitro tests to optimise the diagnosis of these reactions is essential for patients who may be receiving first-line treatment for their cancer.”

  1. Summarize the findings at the end of each section in the Results.

Following your recommendation, we have introduced the following sentences at the end of the Results sections.

  • 2.3. Basophil activation test by flow cytometry (BAT-FC) (line 134):

 “Therefore, the low sensitivity of BAT-FC makes this technique ineffective to classify patients in our study.”

  •  2.4. sIgE microarrays immunoassay (line 167):

 “In conclusion, the sIgE microarray immunoassay showed an excellent performance in the classification of patients with IgE-mediated hypersensitivity reactions to PC.”

  • 2.5. Basophil activation test on microarray support (BAT-microarray) (line 193):

“According to our results, both sIgE and BAT microarray showed excellent performance and could improve the endophenotyping of IgE-mediated hypersensitivity reactions to platinum compounds.” 

  1. Please explain the age of the participants and whether the underlying medical conditions are common.

Following your recommendation, we have added the following sentence in the section of Characteristics of the patients (line 89):

The patients' ages ranged from 34 to 82 years (mean 63 years) and 28 patients were being treated for a malignancy (19 colorectal, five genital, one lung and one breast).

  1. It is recommended that the content format in the table be centered.

We have centered the contents of the tables.

  1. On line 305, what was the basis for choosing these three concentrations?

We have selected these concentrations based on the previous experience published by different groups (Ornelas, C., Caiado, J., et al. (2018) The Contribution of the Basophil Activation Test to the Diagnosis of Hypersensitivity Reactions to Oxaliplatin. International archives of allergy and immunology, 177(3), 274–280. https://doi.org/10.1159/000490313 and Giavina-Bianchi, P., et. al (2017) Basophil Activation Test is a Relevant Biomarker of the Outcome of Rapid Desensitization in Platinum Compounds-Allergy. The journal of allergy and clinical immunology. In practice, 5(3), 728–736. https://doi.org/10.1016/j.jaip.2016.11.006 ). We observed cytotoxicity at the highest concentrations tested in a pilot study. Therefore, we decided not to exceed the 250 ug/mL concentration to avoid toxicity that could interfere with the test result.

Reviewer 2 Report

Comments and Suggestions for Authors

The manuscript „Specific IgE and basophil activation test by microarray: a prom-2 ising tool for diagnosis of platinum compound hypersensitivity 3 reactions” by Fernández-Lozano et al. compares the basophil activation test by flow cytometry (BAT-FC) and the newly developed sIgE-microarray and BAT-microarray in diagnosing IgE-mediated hypersensitivity reactions to platinum-based compounds (PC). This topic is important, and the manuscript is well-written. The experiments seem to be properly performed and evaluated. I have only few comments:

1. What is the difference between RBL-30/25 and RBL SX-38 cell lines?

2. Why in the BAT-FC test a human IL-3 concentration of 9 ng/mL was used?

3. Please check the sentence in the lines 84-87, I think something is missing: “The severity of the initial reaction was moderate or severe (Brown's grade 2-3/ RCUH classification 2-4) in 62% (15/24) and 38% (9/24) of patients suffered a mild initial reaction (Brown's grade 1/RCUH classification 1)”.

Author Response

Dear reviewer,

Thank you for your careful review of our paper and for the valuable comments, corrections, and suggestions that have helped us to improve the manuscript. Below, you will find the point-by-point response to your questions.

  1. What is the difference between RBL-30/25 and RBL SX-38 cell lines?

In order to assess human IgE sensitization, several groups have created stably transfected humanized rat basophilic leukemia cells (RBL). RBL SX-38 cell line express the alpha, beta, and gamma chains of human Fc epsilon RI and RBL-30/25 express only the alpha chains of human Fc epsilon RI. In both cases, the ability to degranulate after cross-linking of human IgE-bound FcεRI has been demonstrated.

  1. Why in the BAT-FC test a human IL-3 concentration of 9 ng/mL was used?

Over the last few years, when we have been performing the basophil activation test as a diagnostic tool for different drugs, we have tested different concentrations of IL-3 and have chosen the final concentration of 9 ng/ml, taking into account that this concentration is optimal to achieve correct degranulation in the basophil activation test. Furthermore, the same concentration is used by other groups in their work, for example “De Vlieger, L., Nuyttens, L., Ieven, T., Diels, M., Coorevits, L., Cremer, J., Schrijvers, R., & Bullens, D. M. (2023). Basophil activation test with progressively less heated forms of egg distinguishes egg allergic from tolerant children. Journal of investigational allergology & clinical immunology https://doi.org/10.18176/jiaci.0964 ”. 

  1. Please check the sentence in the lines 84-87, I think something is missing: “The severity of the initial reaction was moderate or severe (Brown's grade 2-3/ RCUH classification 2-4) in 62% (15/24) and 38% (9/24) of patients suffered a mild initial reaction (Brown's grade 1/RCUH classification 1)”.

We modify the sentence as follows (line 91):

“Fifteen of 24 (62%) patients suffered a moderate or severe initial reaction (corresponding to Brown's grade 2-3 and RCUH grade 2-4), and 9 of 24 (38%) presented a mild initial reaction (corresponding to Brown's grade 1 and RCUH grade 1).”